# Identification of Thermo-Sensitive Chloroplast Development Gene *TSCD5* Required for Rice Chloroplast Development under High Temperature

Shenglong Yang [1,2,†], Guonan Fang [1,†], Banpu Ruan [2], Anpeng Zhang [2], Yun Zhou [1], Guangji Ye [1], Wang Su [1], Heng Guo [1], Jian Wang [1,*] and Zhenyu Gao [2,*]

[1] Key Laboratory of Qinghai-Tibet Plateau Biotechnology Ministry of Education, Qinghai Academy of Agriculture and Forestry Sciences, Qinghai University, Xining 810016, China
[2] China National Rice Research Institute, Hangzhou 310006, China
[*] Correspondence: jianwang2197@163.com (J.W.); gaozhenyu@caas.cn (Z.G.)
[†] These authors contributed equally to this work.

**Abstract:** Chloroplasts are critical organelles for photosynthesis and play significant roles in plant growth and metabolism. High temperature is one of abiotic stresses affecting the growth and development of plants, involving chlorophyll biosynthesis and chloroplast development. It is well known that the methyl erythritol 4-phosphate (MEP) pathway is vital to photosynthesis and plant growth, and 1-deoxy-D-xylulose-5-phosphate synthase (DXS) is the enzyme that catalyze the first step of the MEP pathway. Although DXS has been widely studied in microbes and plants, no DXS gene has been identified in rice. Here, a novel *thermo-sensitive chlorophyll-deficient 5 (tscd5)* mutant was isolated in rice with decreased chlorophyll contents, impaired chloroplasts, and albino leaves at high temperature (35 °C). Fine mapping and DNA sequencing of *TSCD5* found a missense mutation (G to A) in the sixth exon of *LOC_Os05g33840* in *tscd5*. The *TSCD5* gene encodes a 1-deoxy-D-xylulose-5-phosphate synthase 1 (*OsDXS1*) localized in chloroplast. Complementation tests and overexpression assay demonstrated that the mutation in *LOC_Os05g33840* caused the *tscd5* phenotype. qRT-PCR of *TSCD5* showed it was constitutively expressed in all tissues, and its transcript amounts were reduced in *tscd5* under high temperature. Here, *TSCD5* is verified to be crucial to chloroplast development under high temperature in rice, which may facilitate the elucidation of the molecular mechanisms which underlie acclimation to high temperature stress in plants.

**Keywords:** *TSCD5*; chlorophyll biosynthesis; chloroplast development; high temperature; rice





## 1. Introduction

Chloroplasts are essential organelles for photosynthesis in plants [1], which are involved in carbon assimilation, energy transfer and amino acid synthesis [2,3]. Therefore, the normal development of chloroplast is important for maintaining crop yield. The chloroplast is a semi-autonomous organelle encoding approximate 100 genes and 60–80 proteins [4]. Thus, mutation of these chloroplast-relative genes might cause many abnormal leaf mutants. These mutants were not only considered to be ideal materials for uncovering the molecular mechanisms of chlorophyll biosynthesis, chlorophyll degradation, photosynthetic and chloroplast development [5]. So far, many leaf-color mutants were identified with various phenotypes, such as white stripe leaf, albino, yellow green, and green-revertible albino leaf in rice. Interestingly, these mutants usually respond to temperature and light. The *TCM5* gene encodes chloroplast-targeted Deg protease protein, which has been found to play a role in chloroplast development and the maintenance of PSII function [6]. The *WLP2* gene encodes a PEP-associated protein, and protects the development of chloroplast under heat stress in rice through a TRX-FLN regulatory module [7]. The *HSA1* gene encodes a plastid-encoded RNA polymerase that plays a key role in chloroplast development in the early

stages of rice and functions to protect chloroplasts from heat stress in the later stages [8]. *TSCD11* encoding a seryl-tRNA synthetase important to chloroplast development under high temperature were identified in rice [9]. The *V1*, *V2*, *V3*, *TCD5*, *TCD9*, *TCD10*, *TCM12* and *DUA1* are leaf-color mutants at the seedling stage which develop albino leaves at low temperatures [10–17]. In addition, the *ls1* exhibited local lesions phenotype, and induced accumulation of ROS, which caused the leaf cells damage and apoptosis under high light and high temperature conditions [18].

As a primary abiotic stress, high temperature can affect the growth and development of plants, produce irreversible damage and even cause death, which affect the yield and quality of crops [8,19]. For example, when the environmental temperature outstrips to 35 °C, the seed germination, seedling and vegetative growth, flowering and seeding maturity were unfavorably affected [20]. Chloroplasts are extremely sensitive to heat, which affect photochemical reactions, electron transport, photophosphorylation, $CO_2$ assimilation, thylakoid membrane fluidity and chlorophyll biosynthesis [21,22]. Simultaneously, high temperature will directly cause the inactivation of enzymes in chloroplast and mitochondria, the inhibition of protein synthesis, protein degradation, even protein denaturation and aggregation and the loss of membrane integrity [23]. As the chloroplast becomes damaged, the plants tend to become weaker [9]. These weak plants are often sensitive to reactive oxygen species (ROS), such as hydroxyl radicals ($\cdot$OH), superoxide anions ($O_2^-$) and hydrogen peroxide ($H_2O_2$), especially in stress situations [9]. Common stress situations include high light or high temperature. ROS can cause wide damage to DNA, proteins and lipids [24–26]. At the same time, antioxidants and antioxidant enzymes in plants were induced under stress conditions including the glutathione (GSH) and antioxidants ascorbic acid (AsA), and antioxidant enzymes, such as superoxide dismutase (SOD), catalase (CAT) and peroxidase (POD).

The mevalonate (MVA) and MEP pathways are the two pathways which can both produce isoprenoids and are essential for photosynthesis and plant growth [27,28]. The isoprenoids produce multiple compounds in the MVA and MEP pathway, including carotenoids, chlorophylls, tocopherols, plastoquinone, phylloquinone, which regulate the growth of plants [29,30]. DXS is an enzyme that catalyzes the first steps of the MEP pathways, which is vital to bacteria (*E. coli*) and plants [31,32]. The gene was first isolated from *E. coli*, and subsequently its homologous genes were isolated from *Arabidopsis*, *Mentha haplocalyx Briq* and other bacteria [33–36]. The gene family of DXS has been divided into three categories, namely I-DXS, II-DXS and III-DXS. The I-DXS enzymes play significant roles in the biosynthesis of housekeeping and photosynthetic terpenoids such as chlorophylls and carotenoids of leaves in *Arabidopsis*, maize (*Zea mays*), rice and Medicago (*Medicago truncatula*) [35,37–39]. In *Arabidopsis*, the isolation of the *CLA1* gene is required for the both chlorophyll and carotenoids synthesis, and the *CLA1* gene mutation lead to a lack of chlorophyll and carotenoids appeared with an albino phenotype in *cla1* mutant [31,32]. During the vegetative and reproductive stages of rice, the transcript level of *OsDXS1* in leaves was significantly higher than that in other tissues [28]. The II-DXS enzymes play secondary and ecological roles in the generation of functional terpene metabolites, such as carotenoids in yellow kernels of maize, apocarotenoids in mycorrhizal roots of Medicago and phytoalexin in rice [39–41]. *OsDXS2* plays a vital role as a speed limited enzyme providing IPP/DMAPPs to the accumulation of seed-carotenoid in rice [28]. The III-DXS enzymes have indeterminate roles concretely in the *Poaceae* family and a few dicotyledons in angiosperms [38,39,42,43]. Although the function of *DXS* has been reported in bacteria and plants, few studies have reported its role in chloroplast development in rice under high temperature.

In our study, a new rice *thermo-sensitive chlorophyll-deficient 5* (*tscd5*) mutant was isolated from ethyl methanesulfonate (EMS) of *Oryza sativa japonica cv.* Wuyunjing 7 (WYJ7), which exhibited albino leaf phenotype under high temperature. Genetic complementation and overexpression assays demonstrated that *tscd5* point mutation was the cause of the

phenotype of thermo-sensitive albino. *TSCD5* encodes a chloroplast-targeted DXS1 which is essential for the chloroplast development under high temperature.

## 2. Materials and Methods

### 2.1. Plant Materials and Growth Conditions

The rice leaf color mutant, *tscd5*, was isolated from *japonica* var. Wuyunjing 7 (WYJ7) treated with ethyl methane sulfonate (EMS). The *tscd5* mutant was crossed with its wild-type WYJ7 and an *indica* variety 93–11, respectively to generate populations for genetic analysis and fine mapping. These plants were all grown in the paddy field in Fuyang, Zhejiang province (119°95′ E, 30°05′ N). For temperature treatment experiments, the WYJ7 and *tscd5* plants were cultured in growth chambers (light for 12 h and dark for 12 h; light intensity 200 µmol m$^{-2}$ s$^{-1}$) with a sustained temperature of either 25, 30, and 35 °C. For the shift lower-temperature (25 °C) to higher-temperature (35 °C) and shift higher-temperature (35 °C) to lower-temperature (25 °C) temperature conversion experiments, the seeds of WYJ7 and *tscd5* were cultured in a incubators (light for 12 h and dark for 12 h; light intensity of 200 µmol m$^{-2}$ s$^{-1}$) at the sustained temperature (25 °C or 35 °C), respectively, reached the 1-leaf or 2-leaf stage, and then cultured in an incubator (light for 12 h and dark for 12 h; light intensity of 200 µmol m$^{-2}$ s$^{-1}$) at the sustained temperature (35 °C or 25 °C) reached the 3-leaf stage.

### 2.2. Measurement of Chlorophyll Content

The chlorophyll was extracted from different growth periods of fresh leaf samples (0.1 g) of WYJ7 and *tscd5* plants in the paddy field and at 25 °C and 35 °C in growth chambers. The plants were cut into small segments, then put them into 10 mL of 80% acetone, soaked in darkness for 24 h, and shaken every 6 h to 8 h. Ultraviolet spectrophotometer (DU800, BECKMAN, Fullerton, USA) was used to measure chlorophyll at 470, 645 and 663 nm. Each condition was analyzed with three biological replicates.

### 2.3. Measurement of Photosynthetic Rate

According to a method described by Li et al. [44], the LI-6800 portable photosynthetic tester manufactured by LI-COR of the United States is used with 28 °C, irradiance of 1200 µmol photons m$^{-2}$ s$^{-1}$ and $CO_2$ concentration of 400 mol mol$^{-1}$ in the paddy field during the tillering stage. The leaves of WYJ7 and *tscd5* plants were sampled at the tillering stage to determine their net photosynthetic rate (*P*n). Each condition was analyzed with three biological replicates.

### 2.4. Transmission Electron Microscopy (TEM)

The fresh leaves were sampled from wild-type and *tscd5* mutant at seedling in growth chambers and at tillering stage in the field conditions, cut into small segments, and containing 2.5% glutaraldehyde fixative for > 4 h. According the method of Li et al. [44], all samples were subjected to 1%(w/v) $OsO_4$ in phosphate, dehydration, infiltration embedding, dicing, using uranyl acetate and alkaline lead citrate, staining and observed using a transmission electron microscope (Hitachi Model H-7650).

### 2.5. TUNEL Assay

The TUNEL assay is a method to detect DNA fragmentation that arises from apoptotic signaling cascades through the action of endogenous endonucleases by labeling the terminal end of nucleic acids [45,46]. The fresh leaves were cut into small pieces and fixed with FAA fixative, and then embedded in paraffin. The sections were inspected to choose the proper slide and dewaxed with gradient alcohol. All following TUNEL staining steps were from the method of the Huang et al. and Ruan et al. [47,48], and apoptosis was detected by Promega kit from USA. Samples of apoptotic cells (fluorescein-12-dUTP) were examined by laser scanning confocal microscopy (Zeiss LSM700, Carl Zeiss, Inc., NY, USA) for localized green fluorescence (520 nm) against a red (620 nm) background (PI, propidium iodide).

### 2.6. Nitro Blue Tetrazolium (NBT) and 3, 3′-Diaminobenzidine (DAB) Staining

Detection of $O_2^-$ and $H_2O_2$ were conducted by NBT and DAB staining [49,50]. Fresh leaves of WYJ7 and *tscd5* were obtained from temperature-treated seedlings at 3 leaf stages, and samples were immersed in 0.05% (w/v) NBT or 0.1% (w/v) DAB (pH 5.8) for 12 h at 28 °C in the dark with gentle shaking.

### 2.7. Measurement of Physiological Indices Related to ROS

The malondialdehyde (MDA) and $H_2O_2$ contents, as well as SOD, CAT and POD activities were determined using the Assay Kit (Suzhou Keming Biotechnology Co, Ltd. In, Suzhou, Jiangsu, China). Leaves of WYJ7 and *tscd5* were sampled at the 3-leaf stage from seedlings cultured in the temperature treatments of 25 °C and 35 °C. Three biological replicates were utilized, and then *t*-test was performed.

### 2.8. Genetic Analysis and Fine Mapping

Genetic analysis was conducted by $F_2$ population crossed with *tscd5* mutant and the *japonica* WYJ7, and confirmed the single recessive gene controlled the *tscd5* phenotype. The *tscd5* mutant and the 93–11 (*indica* rice cultivar) were crossed to generate $F_2$ population for fine mapping. For the initial localization of *TSCD5*, 94 individual plants of the *tscd5* mutant phenotype were selected. In total, 225 SSR markers distributed on 12 rice chromosomes (www.gramene. org) were used for preliminary localization. Then, 376 plants of the tscd5 mutant phenotype in $F_2$ populations were chosen for fine mapping. Six indel and one SSR marker between the B5-10 and B5-12 were developed based on genomic DNA sequences distinction in Nippobare (*japonica* rice variety) and the 93–11 (*indica* rice variety) to shrink the *TSCD5* interval. The primers utilized are listed in Supplementary Table S3.

### 2.9. Sequence and Phylogenetic Analysis

The TSCD5 protein sequence containing 295 amino acids were acquired from The National Center for Biotechnology Information database (http://www.ncbi.nlm.nih.gov/BLAST/, accessed on 18 May 2022). Protein sequence alignment was performed by DNA-MAN. The phylogenetic tree of TSCD5 and homologous proteins was constructed by MEGA 7.0 software according the neighbor-joining method with 1000 replicates.

### 2.10. Plasmid Construction and Rice Transformation

For complementation experiment, a 7642 bp genomic DNA fragment of *TSCD5* encompasses the native promoter, coding sequence and terminator regions, which was cloned into the pCAMBIA 1300 vector. So as to construct an overexpression vector, 2163 bp coding sequence of *TSCD5* was cloned into pCAMIA1300S with 2 × 35S promoter. The vectors of complementation and overexpression were introduced into calli of the *tscd5* mutant via an *Agrobacterium* (EHA105)-mediated approach. The primers utilized are listed in Supplementary Table S4.

### 2.11. Subcellular Localization of TSCD5 Protein

The full-length coding region of *tscd5* was amplified by PCR, digested with Sal*I*, and cloned into a pCA1301-35S-S65T-GFP vector to generate the p35S-TSCD5-GFP fusion construct. The construct of p35S-TSCD5-GFP was introduced into rice protoplasts and transiently expressed. GFP signals were detected by laser scanning confocal microscopy (Zeiss LSM 700, Oberkohen, Germany). The primers utilized are listed in Supplementary Table S4.

### 2.12. GUS Assay

The promoter of *tscd5* was amplified from WYJ7 genomic DNA by PCR, digested with BamHI and NocI, which was then inserted into a pCAMBIA 1305 vector with the GUS reporter gene. All vectors were introduced into the callus of WYJ7 via *Agrobacterium* (EHA105)-mediated approach. GUS determination was performed in roots, stems, leaves, sheaths and panicles of transgenic plants. The sequences of primers used are listed in

Supplementary Table S4. For GUS staining, independent T1 transgenic plants of excised tissues were employed according to method described by Rao et al. [51].

### 2.13. RNA Extraction and Quantitative Real-Time PCR (qRT-PCR) Analysis

Total RNA was extracted from varied organs as previous description [52]. The cDNA was reverse-transcribed with ReverTra Ace qPCR-RT Master Mix (Toyobo, Japan). The experiment of qRT-PCR was conducted with SYBR premix Ex Taq II (Takala, Japan) in the CFX96 Touch$^{TM}$ Real-Time PCR (Bio-Rad, California, CA, USA) by the following manufacturer's instructions. All target genes were examined for related expression levels choosing the *Histone* gene (*LOC_Os06g04030*) as the internal reference [9]. The real-time PCR experiments were conducted with three biological replicates and the *t*-test was employed for statistical analysis. The primers utilized in qRT-PCR experiment are listed in Supplementary Table S5.

### 2.14. Statistical Analysis

The significance analysis of data from our experiments was conducted using *t*-test in Excel software. * significance at $p < 0.05$, and ** extremely significance at $p < 0.01$ (Student's *t*-test).

## 3. Results

### 3.1. The Mutant Phenotype of tscd5 Is Sensitive to High Temperature

The *tscd5* mutant, derived from EMS-treated *japonica* rice cv. Wuyunjing 7 (WYJ7), displayed distinct phenotypes from WYJ7 when sown in June or July in Hangzhou. In the paddy field of Hangzhou, the *tscd5* mutant seedlings exhibited local albino leaf phenotype at an average temperature of 30 °C in June (Figure 1A), while it exhibited albino leaf phenotype at an average temperature of 35 °C (high temperature) in July (Figure 1B). In addition, the *tscd5* mutant showed local albino on leaves at the tillering and heading stage at an average temperature of 30 °C in field conditions (Figure 1C,D). Compared with WYJ7, the *tscd5* mutant showed significantly lower chlorophyll content and net photosynthetic rate (Pn) at the tillering stage, respectively (Figure 1H,I). Moreover, agronomic traits, such as plant height, panicle length, number of secondary branches of panicle, grain width, 1000-grain weight and seed setting rate of the *tscd5* plants were much lower than those of WYJ7 plants at high temperature in the field (Supplementary Table S1).

When grown in growth chambers at the three-leaf stage, the *tscd5* seedlings displayed normal phenotype with green leaf at 25 °C (Figure 1E), partial albino leaf at 30 °C (Figure 1F), and albino leaf phenotype and then gradually died at 35 °C (Figure 1G). Pigment content determination indicated that chlorophyll *a* (Chl *a*), chlorophyll *b* (Chl *b*) and carotenoid (Car) contents of the *tscd5* plants were similar to WYJ7 plants at 25 °C (Figure 1J), while significantly lower than WYJ7 plants at 30 °C and 35 °C (Figure 1K,L). Since the *tscd5* mutant exhibited albino phenotype from one-leaf stage, the shift-temperature experiment were conducted at one-leaf stage and two-leaf stage, respectively (Figure 2A–H). When transferred from 25 °C to 35 °C during the one-leaf stage, the *tscd5* mutant generated new second leaf with albino phenotype (Figure 2A,E), and Chl *a*, Chl *b* and Car contents of *tscd5* were just 2.45%, 5.29% and 2.84% of WYJ7, respectively (Figure 2I). When moved from 25 °C to 35 °C during the two-leaf stage, the new third leaf of the *tscd5* mutant also displayed albino phenotype (Figure 2B,F), and Chl *a*, Chl *b* and Car contents were just 2.68%, 4.80% and 4.22%, respectively of WYJ7 (Figure 2J). On the contrary, whether transferred from 35 °C to 25 °C during the one-leaf stage or during the two-leaf stage, new leaves were restored to normal green as WYJ7 (Figure 2C,D,G,H,K,L). Therefore, pigment synthesis in new developing leaves of the *tscd5* mutant is temperature-dependent.

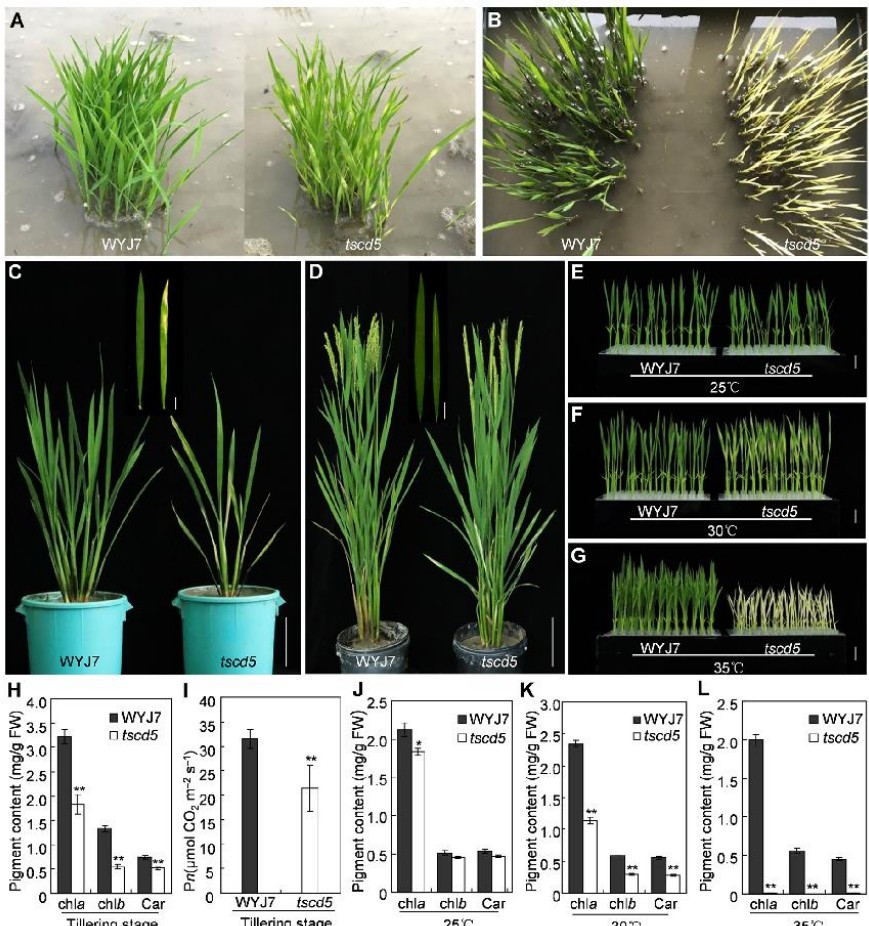

**Figure 1.** Phenotypic characterization of the WYJ7 and *tscd5* mutants in rice. (**A**) Phenotypes of WYJ7 and *tscd5* grown in the June at seedling stage in the paddy field. (**B**) Phenotypes of WYJ7 and *tscd5* grown in the July at seedling stage in the paddy field. (**C**) Phenotype of WYJ7 (left) and *tscd5* mutant (right) at the tillering stage grown in the paddy field. (**D**) Phenotype of WYJ7 (left) and *tscd5* mutant (right) at the heading stage grown in the paddy field (Scale bar = 10 cm). (**E–G**) Phenotype of WYJ7 (left) and *tscd5* mutant (right) grown at 25 °C (**E**), 30 °C (**F**), 35 °C (**G**), respectively (Scale bar = 2 cm). (**H**) Pigment contents of the tillering stage of WYJ7 and *tscd5* mutants grown in the paddy field. (**I**) Net photosynthetic rate ($P_n$) of the tillering stage of WYJ7 and *tscd5* mutants grown in the paddy field. (**J–L**) Pigment content of the three-leaf stage of WYJ7 and *tscd5* mutants grown at 25 °C (**J**), 30 °C (**K**), 35 °C (**L**), respectively. Mean ± SD, n = 3, * significance at $p < 0.05$, ** extremely significance at $p < 0.01$ (Student's *t*-test).

### 3.2. Chloroplast Development Is Hampered and Cell Death Is Exacerbated in tscd5 at High Temperature

The ultra-structure of chloroplast in the mesophyll cells of WYJ7 and *tscd5* plants at the three-leaf stage were visualized by transmission electron microscopy (TEM) (Figure 3A–H). At 25 °C, the chloroplast of WYJ7 and *tscd5* plants exhibited well-developed lamellar structures furnished with normally stacked grana and thylakoid membranes (Figure 3A,B,E,F). However, the albino leaf cells in *tscd5* contained fewer chloroplasts and grana lamellae compared with those in WYJ7 at 35 °C (Figure 3C,D,G,H). TEM observation also found that thylakoids and stroma lamellae structure in leaves were disorderly arranged and more osmiophilic granules (OG) in the *tscd5* mutant compared to WYJ7 during the tillering stage at high temperature (Supplementary Figure S1). Therefore, chloroplast development was hampered in the *tscd5* mutant at high temperature.

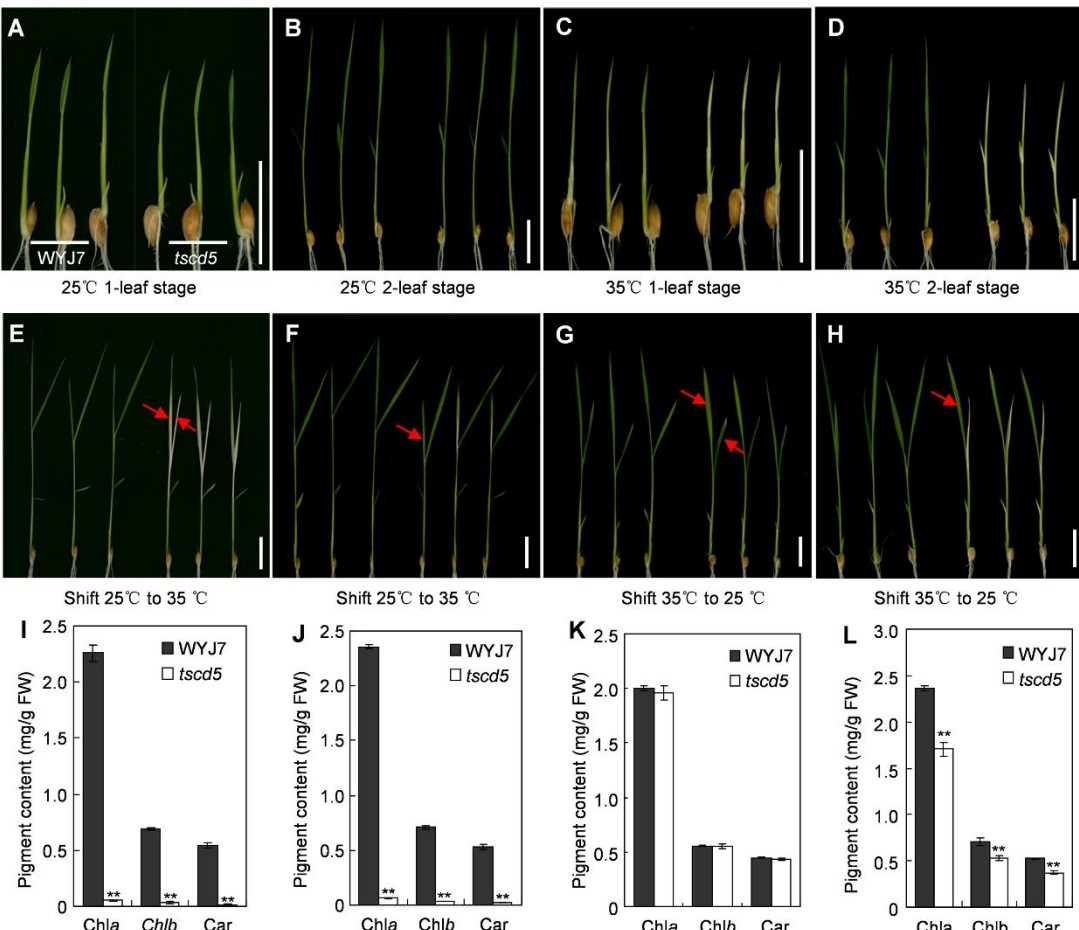

**Figure 2.** Phenotypes and pigment contents comparison between WYJ7 and *tscd5* plants at shift 25 °C to 35 °C and shift 35 °C to 25 °C in rice. (**A–D**) Phenotypes of the WYJ7 and *tscd5* at the one-leaf and two-leaf stage grown at continuous 25 °C and 35 °C, respectively. (**E–H**) Phenotypes of the WYJ7 and *tscd5* at the three-leaf stage grown at continuous 25 °C shift 25 °C to 35 °C, continuous 35 °C, shift 35 °C to 25 °C, respectively. (**I–L**) Pigment contents of the new third-leaf of WYJ7 and *tscd5* plants grown at first-leaf shift 25 °C to 35 °C (**I**), second-leaf shift 25 °C to 35 °C (**J**), first-leaf shift 35 °C to 25 °C (**K**), second-leaf shift 35 °C to 25 °C (**L**). (Scale bar = 2 cm). Mean ± SD, n = 3. ** extremely significance at *p* < 0.01 (Student's *t*-test).

To determine cell death in the *tscd5* mutant, the third leaf tissues of WYJ7 and *tscd5* plants grown at 25 °C and 35 °C at the three-leaf stage were subjected to a TUNEL assay (Figure 3I–P). The sections of third leaf were simultaneously stained with PI to show the nuclei (red) in each section. In the third leaf sections of WYJ7 and *tscd5* grown at 25 °C, few nuclei were TUNEL positive (Figure 3I,J,M,N); whereas at 35 °C, more nuclei in *tscd5* were found TUNEL positive than in WYJ7 (Figure 3K,L,O,P). Additionally, transcriptional expression of senescence-associated genes (SAGs), including *Osh36*, *Osl57*, *Osl85*, *OsWRKY23*, *OsWRKY72*, *OsNAC2* and *SGR* were highly induced in *tscd5* at the three-leaf stage at 35 °C (Supplementary Figure S2A,C).

### 3.3. Excess ROS Accumulation in tscd5 at High Temperature

High temperature stress usually induces the accumulation of ROS in plant cells [53]. The noxious ROS can further lead to lipid peroxidation, cellular damage and even cell death. We used NBT and DAB staining to detect $H_2O_2$ and $O_2^-$, respectively. The fresh leaves of the WYJ7 and *tscd5* plants cultured at 25 °C and 35 °C at the three-leaf stage were incubated in NBT and DAB. NBT and DAB staining showed that the copious $H_2O_2$ and $O_2^-$ accumulation were visualized in leaves of *tscd5* cultured at 35 °C (Figure 4A,B,H,I).

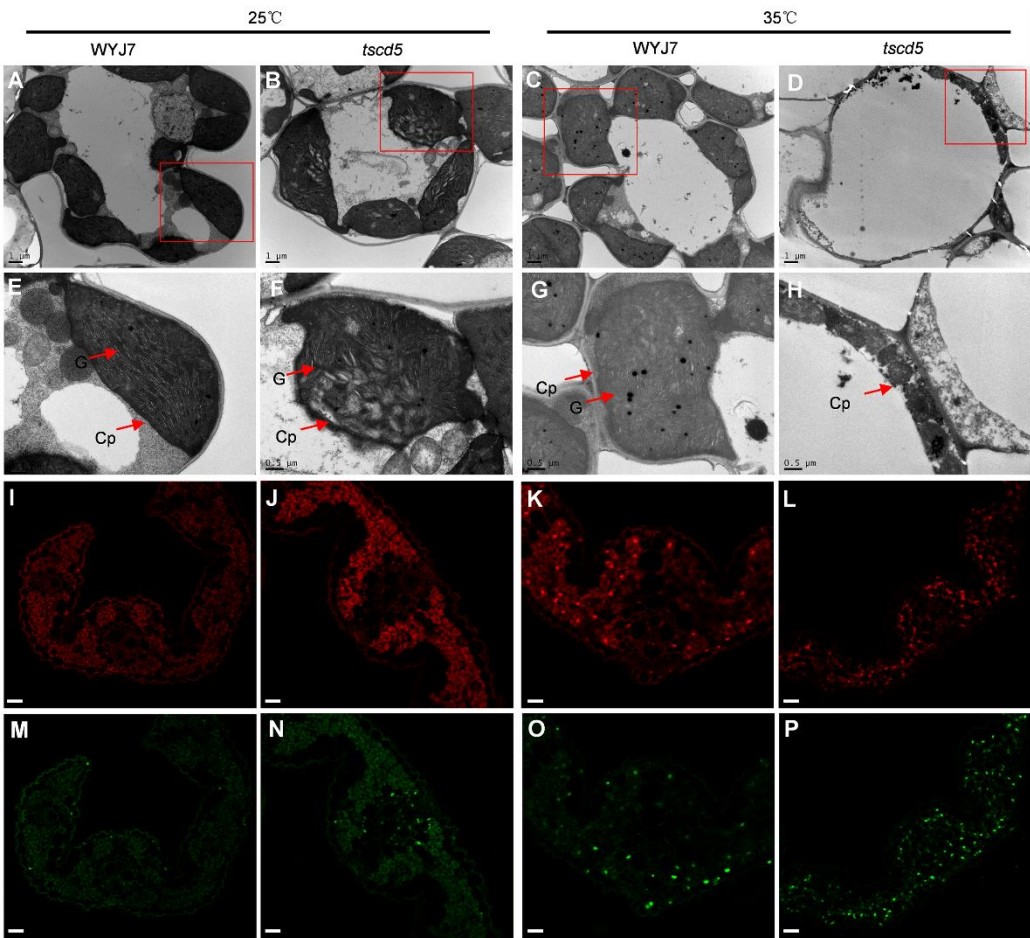

**Figure 3.** Transmission electron micrographs of chloroplasts and TUNEL assay in WYJ7 and the *tscd5* leaves of rice. (**A**,**B**) Chloroplasts structure from WYJ7 and *tscd5* at 25 °C (Scale bar = 1 m). (**C**,**D**) Chloroplasts structure from WYJ7 and *tscd5* at 35 °C (Scale bar = 1 μm). (**E–H**) are enlargements of a-d, respectively (Scale bar = 0.5 μm). CP, chloroplast; G, grana lamella stacks. (I-P) TUNEL assay of leaves. RED signal is DAPI staining, green color represents positive result (Scale bar = 50 μm). (**I**,**J**,**M**,**N**) leaves from WYJ7 (**I**,**M**) and *tscd5* (**J**,**N**) plants at 25 °C. (**K**,**L**,**O**,**P**) leaves from WT (**K**,**O**) and *tscd5* (**L**,**P**) plants at 35 °C.

The content of senescence-related substance, encompassing $H_2O_2$ and MDA, and CAT, SOD and POD activities were detected in leaves of the WYJ7 and *tscd5* plants cultured at 25 °C and 35 °C. At 25 °C, the $H_2O_2$ and MDA contents, and CAT, SOD and POD activities in the *tscd5* mutant showed no significant distinctions compared to WYJ7 (Figure 4C–G). However, at 35 °C, $H_2O_2$ and MDA contents, and the activities of SOD, POD significantly increased (Figure 4J,K,M,N), and the activity of CAT significantly decreased in the *tscd5* mutant compared to WYJ7 (Figure 4L). Accordingly, transcriptional expression levels of several ROS-related genes, such as *Alternative Oxidase* (*AOX1a*, *AOX1b*), *Ascorbate Peroxidase* (*APX1*), *Catalase* (*CATB*) were significantly up-regulated, but *Ascorbate Peroxidase* (*APX2*), *Catalase* (*CATA*) were remarkably down-regulated in *tscd5* in comparison with WYJ7 at the three-leaf stage at 35 °C (Supplementary Figure S2C,D). These results indicated that excess ROS accumulation in *tscd5* under high temperature stress.

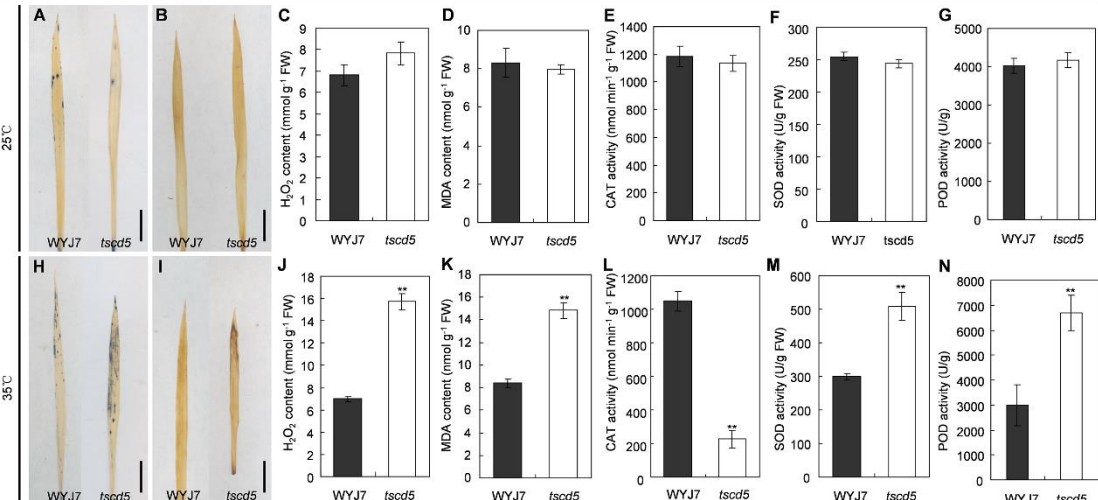

**Figure 4.** Determination of reactive oxygen species (ROS) contents in the WYJ7 and *tscd5* mutant in rice. (**A,H**) NBT staining of leaves between the WYJ7 and *tscd5* plants at 25 °C and 35 °C (Scale bar = 2 cm). (**B,I**) DAB staining of leaves between the WYJ7 and *tscd5* plants at 25 °C and 35 °C (Scale bar = 2 cm). Statistical analysis of $H_2O_2$ contents (**C,J**), MDA contents (**D,K**), and activity of CAT (**E,L**), SOD (**F,M**), POD (**G,N**) at 25 °C and 35 °C. Mean ± SD, n = 3. ** extremely significance at $p < 0.01$ (Student's *t*-test).

### 3.4. Map-Based Cloning of TSCD5

To genetically analyze the *tscd5* mutant, we crossed it with its wild-type WYJ7. All $F_1$ individuals showed the wild-type phenotype and 227 out of 921 plants in the $F_2$ segregating population exhibited the albino phenotype. The resulting 3:1 ($\chi^2 = 1.7361$) segregation ratio indicated that the *tscd5* mutant phenotype is controlled by a single recessive gene (Supplementary Table S2).

For fine-mapping of the *TSCD5* gene, the *tscd5* mutant was crossed 93-11 (*indica* variety) to acquire a $F_2$ segregating population. We first mapped the *TSCD5* locus between two markers, B5-10 and B5-12 on chromosome 5 using 94 $F_2$ individuals with the *tscd5* phenotype (Figure 5A). To fine map *TSCD5*, one SSR and five InDel markers were developed between B5-10 and B5-12. The *TSCD5* locus was finally narrowed to 51.72 kb between ZL-8 and RM440 markers with 376 $F_2$ mutant individuals (Figure 5B). Five open reading frames (ORFs) predicted in this region according to Nipponbare genome (https://rapdb.dna.affrc.go.jp/ accessed on 15 October 2021) (Figure 5C) were then sequenced and only a single nucleotide mutation (G→A) in *LOC_Os05g33840* was detected at 1363 bp from the ATG start codon, which caused an amino acid substitution of $Gly_{455}$ with $Ser_{455}$ (Figure 5D,E).

To verify the identity of *TSCD5*, a complementation experiment was performed and all transgenic plants with pCAMBIA1300:*TSCD5* restored to the wild-type phenotype at high temperature (Figure 5F). As anticipated, gene expression level of *TSCD5* and pigment contents were restored to those of the wild-type at high temperature (Figure 5G,H). An overexpression test also displayed that all transgenic plants returned to the wild-type phenotype (Figure 6A,B). Transcriptional expression level of *TSCD5* was higher than WYJ7 and *tscd5*, and pigment contents were restored to those of WYJ7 at high temperature (Figure 6C,D). Thus, the results confirmed that *LOC_Os05g33840* is *TSCD5*.

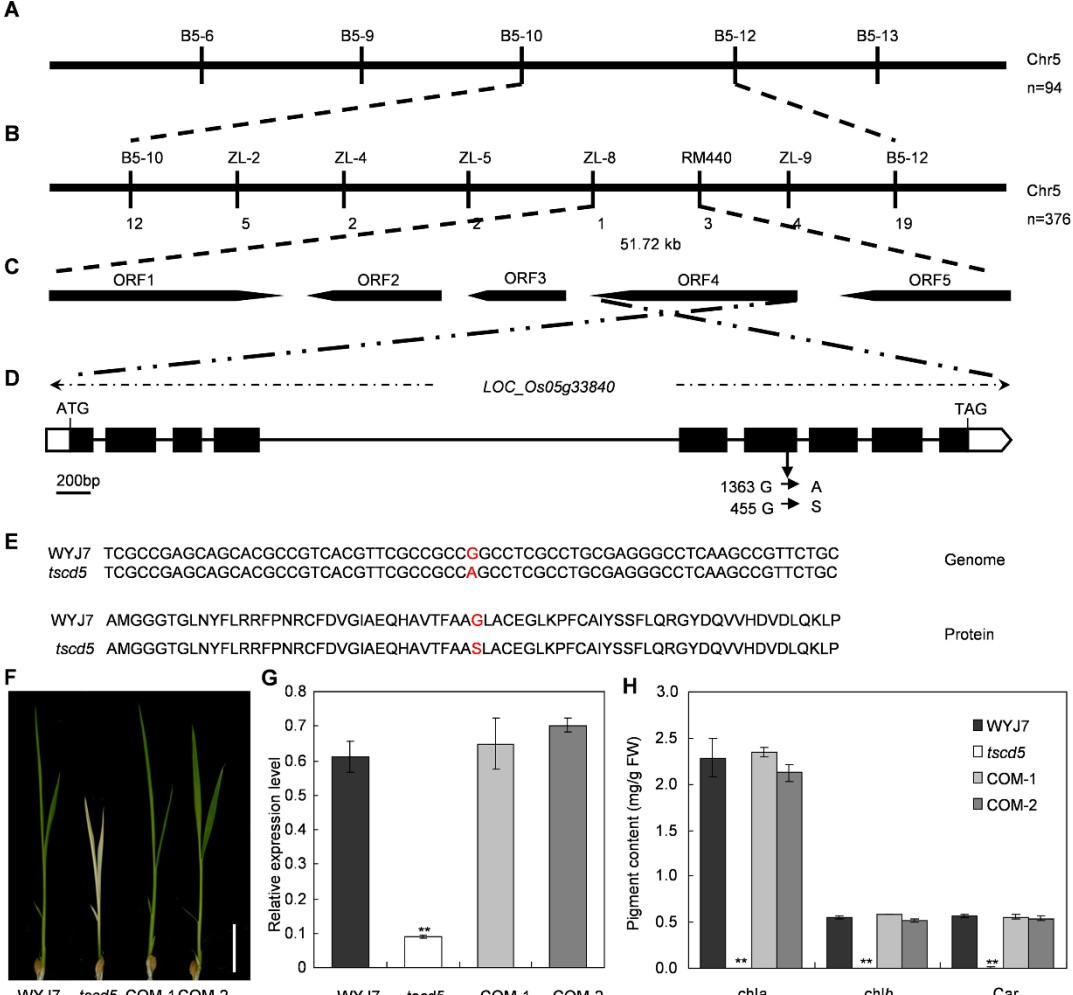

**Figure 5.** Map-based cloning of *TSCD5* and transgenic complementation of the *tscd5* mutant in rice. (**A**) *TSCD5* was preliminarily mapped between markers B5-10 and B5-12 on chromosome 5. (**B**) Fine mapping of *TSCD5*. The *TSCD5* locus was fine mapped to a 51.72 kb region between markers ZL8 and RM440. (**C**) Five putative ORFs are located in the 51.72 kb region. (**D**) Gene structure of the candidate gene *LOC_Os05g33840*. Scale bar =200 bp. The black rectangle represents the exon. The point mutation of G to A on the exon led to amino acid replacement of Gly (G) with Ser (S). (**E**) Comparison of sequence of genome and protein levels around the mutated site between the wild-type WYJ7 and tscd5. (**F**) Functional complementation of *TSCD5* restored phenotypes of the *tscd5* mutant. (**G**) The expression levels of *TSCD5* in WYJ7, *tscd5*, COM-1 and COM-2 at the three-leaf stage grown at 35 °C, *Histone* was used as a control for qRT-PCR. (**H**) Pigment content of the three-leaf stage of WYJ7, *tscd5*, COM-1 and COM-2 plants grown at 35 °C. Mean ± SD, n = 3, ** extremely significance at $p < 0.01$ (Student's *t*-test).

### 3.5. TSCD5 Proteins Are Highly Conserved in Plants

The *TSCD5* gene, encoding a 1-deoxy-D-xylulose-5-phosphate synthase 1 with 497 amino acids (approximately 56 kD), is comprised of nine exons. The TSCD5 protein sequences of different species were compared on NCBI website (http://www.ncbi.nlm.nih.gov/) and a phylogenetic tree was constructed to study the evolutionary relationship between these TSCD5 homologs. Protein sequences alignment showed that TSCD5 were highly conserved in many plant species (Supplementary Figure S3A). Phylogenetic analysis showed that rice TSCD5 had high similarity to the orthologs in *Panicum miliaceum* (91.26%), *Setaria italic* (92.24%), *Zea mays* (91.41%) and *Sorghum bicolor* (90.18%) (Supplementary Figure S3B).

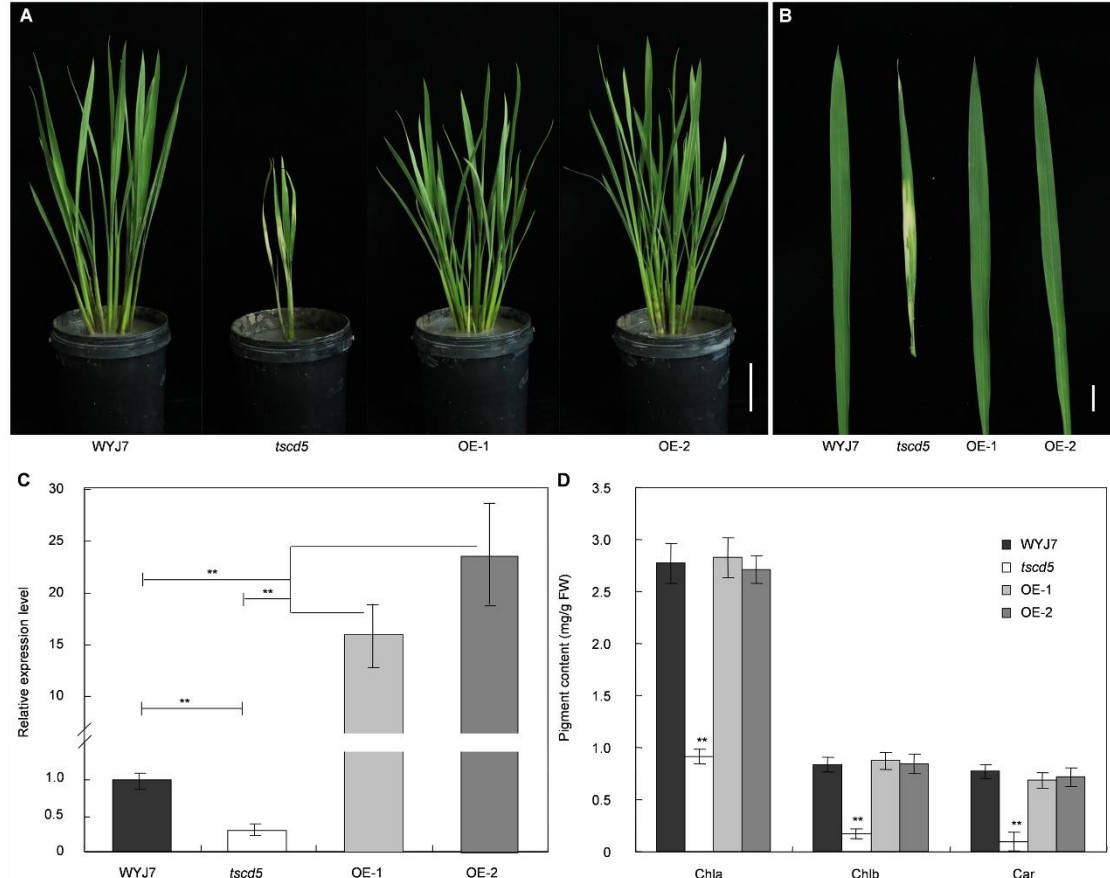

**Figure 6.** Overexpression analysis of *TSCD5* gene in *tscd5* mutant of rice. (**A**) Phenotype of WT, tscd5, OE-1 and OE-2 plants at tillering stage (Scale bar = 10 cm). (**B**) Close-up image of leaves of WYJ7, *tscd5*, OE-1 and OE-2 plants at tillering stage (Scale bar = 2 cm). (**C**) The expression levels of *TSCD5* in WYJ7, *tscd5*, OE-1 and OE-2 at tillering stage, *Histone* was used as a control for qRT-PCR. (**D**) Pigment content of WYJ7, *tscd5*, OE-1 and OE-2 plants at tillering stage. Mean ± SD, n = 3, ** extremely significance at $p < 0.01$ (Student's *t*-test).

### 3.6. TSCD5 Protein Localized in Chloroplasts in Rice

To examine the subcellular localization of rice TSCD5 protein, a transient expression system was constructed by respectively introducing the p35S-GFP and p35S-TSCD5-GFP construct into rice protoplasts. The fluorescence confocal observation revealed that green fluorescence signals for p35S-GFP construct was present in cytoplasm and nucleus (Figure 7A), and signals for p35S-TSCD5-GFP construct were localized to chloroplasts (Figure 7B). The result confirmed that TSCD5 protein located in chloroplasts in rice.

### 3.7. TSCD5 Is Ubiquitously Expressed in Rice Tissues and Significantly Lower in tscd5 Leaves Than That in WYJ7 Leaves under HIGH Temperature

To examine the expression pattern of *TSCD5*, β-glucuronidase (GUS) staining was used to investigate *TSCD5* expression in various tissues. GUS staining indicated that *TSCD5* were widely expressed in stems (S), leaves (L), sheaths (SH), panicles (P) and roots (R), with strongest expression in leaves (Figure 7C–G), which was consistent with RT-qPCR results (Figure 7H).

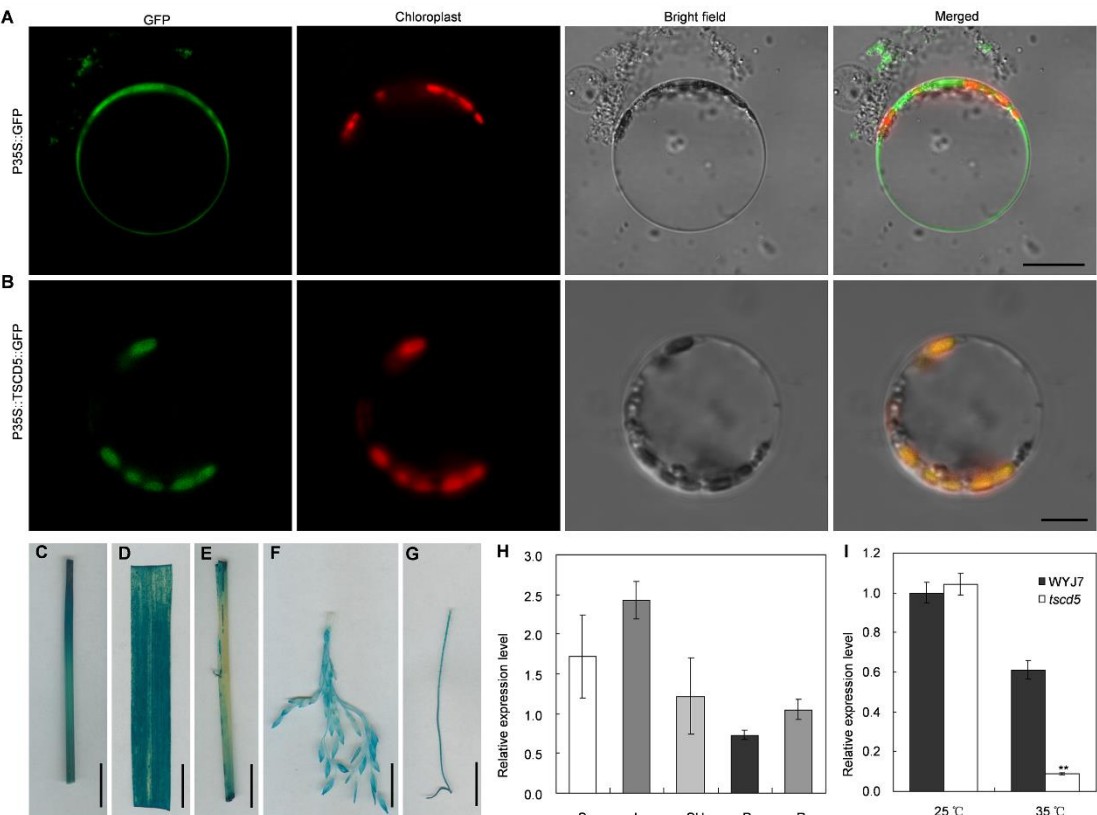

**Figure 7.** Subcellular localization of the TSCD5-GFP protein in rice protoplasts and tissue expression pattern of *TSCD5*. (**A**) Rice protoplast transformed with p35S::GFP as a control. Scale bar = 5 μm; (**B**) Rice protoplast transformed with p35S::TSCD5::GFP. Scale bar = 5 μm. (**C–G**) GUS staining of transgenic rice with p*TSCD5*::GUS at the heading stage. (**C**) Stem. (**D**) Leaf. (**E**) Sheath. (**F**) Panicle. (**G**) Root. Scale bar = 2 cm. (**H**) Relative expression levels of *TSCD5* in various tissues revealed by qRT-PCR. (**I**) Relative expression levels of *TSCD5* in WYJ7 and *tscd5* at the three-leaf stage grown at 25 °C and 35 °C. *Histone* was used as a control for qRT-PCR. Mean ± SD, n = 3, ** extremely significance at $p < 0.01$ (Student's *t*-test).

By RT-qPCR, we also found that the transcriptional level of *TSCD5* was significantly reduced in *tscd5* compared with WYJ7 at high temperature in field (Figure 6C). To ulteriorly plumb whether the mutation and temperature influence *TSCD5* expression, we examined transcriptional levels of *TSCD5* in three-leaf seedlings of *tscd5* and WYJ7 cultured at 25 °C and 35 °C. At 25 °C, *TSCD5* expression levels showed no significant distinction between WYJ7 and *tscd5*. However, its levels in WYJ7 were over six-fold higher than the *tscd5* mutants at 35 °C (Figure 7I).

*3.8. Expression of Genes Related to Chlorophyll Synthesis and Degradation, Chloroplast Development and Photosynthesis Are Affected Dramatically in tscd5 under High Temperature*

We investigated the expression levels of genes associated with chlorophyll synthesis and degradation, chloroplast development and photosynthesis in the *tscd5* mutant and the wild-type. At 25 °C, the expression levels of many tested genes in *tscd5* were mostly similar to those in WYJ7 (Figure 8A–D). While at 35 °C, the expression levels of these tested genes related to chlorophyll biosynthesis, such as *glutamyl tRNA reductase* (*HEMA*), *GSA aminotransferase* (*GSA*), *Heme Oxygenase 1* (*Heme1*), *magnesium-chelatase subunit chlD* (*CHLD*), *divinyl reductase* (*DVR*), *Mg-chelatase H subunit* (*CHLH*), *protochlorophyllide oxidoreductase A/B* (*PORA, PORB*) and *chlorophyll a oxygenase* (*CAO1*) were significantly decreased in *tscd5* compared with WYJ7 plants (Figure 8E); the expression levels of all tested genes for chlorophyll degradation, such as *NON-YELLOW COLORING 1* (*NYC1*), *NON-YELLOW COLORING 3* (*NYC3*), *Red Chlorophyll Catabolite Reductase 1* (*RCCR1*), *PCCR*, *Chlorophyllide a Oxygenase*

(*PAO*) were significantly increased in *tscd5* mutant compared with WYJ7 plants (Figure 8F), which were consistent with the decrease in Chl contents and the albino phenotype in *tscd5* (Figure 1E,G). Compared with WYJ7, expression levels of genes associated with chloroplast development, encompassing *Heat-stress Sensitive Albino 1* (*HSA1*), *Plastid RNA Polymerase* (*OsRpoTp*), *plastidal DNA polymerase* (*OsPOLP1*), *RNA polymerase α, β and β' subunits* (*RpoA*, *RpoB*, *RpoC1*, *RpoC2*), *ribosomal proteins* (*rps7*, *rps15*), *Virescent1* (*V1*), *Virescent2* (*V2*), *Thermosensitive Chlorophyll-deficient mutant 5* (*TCM5*), and photosynthesis related genes, such as *Rubisco Large Subunit* (*RbcL*), *small subunit of Rubisco* (*RbcS*), *Photosystem I Subunit A* (*psaA*), *Photosystem II Subunit of A* (*psbA*), *Chlorophyll a/b Binding Protein* (*CAB1R*, *CAB2R*), *LchP2*, *Lhcb1*, *Lhcb4*, were decreased significantly in *tscd5* (Figure 8G,H).

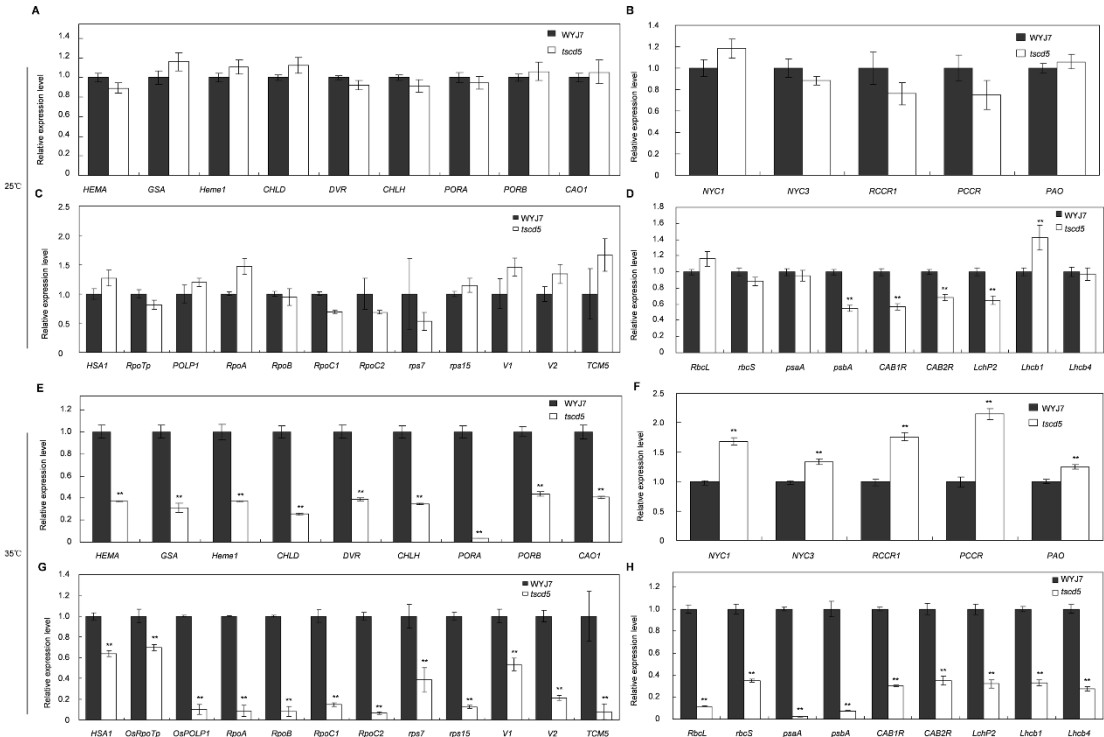

**Figure 8.** Alteration in expression level of chlorophyll synthesis and degradation, chloroplast development and photosynthesis related genes in *tscd5* of rice. (A-D) Expression of genes associated with Chl biosynthesis (**A**) degradation (**B**) chloroplast development (**C**) and photosynthesis (**D**) in WYJ7 and *tscd5* mutant at 25 °C. (**E–H**) Expression of genes associated with Chl biosynthesis (**E**) degradation (**F**) chloroplast development (**G**) and photosynthesis (**H**) in WYJ7 and *tscd5* mutant at 35 °C. *Histone* was used as a control for qRT-PCR. The expression level of each tested genes in WYJ7 was set to 1.0. Mean ± SD, n = 3, ** extreme significance at $p < 0.01$ (Student's *t*-test).

## 4. Discussion

As an important photosynthetic organelle, the chloroplast contains about 100 genes, and more than 3000 proteins within it [54]. In rice, over 80 chlorophyll deficient mutants have been identified so far [55]. Some rice mutants in leaf color showed abnormal phenotypes when the temperature changed [15,56]. In higher plants, 1-deoxy-D-xylulose-5-phosphate synthase (DXS) has been reported involved in multiple biological functions. For example, in Arabidopsis, mutant *cla1/chs5* exhibited normal green leaves at a permissive temperature (22 °C), but chlorotic leaves at low temperature (15 °C). Three *DXS* (*OsDXS1*, *OsDXS2* and *OsDXS3*) genes were reported co-expressed in rice [38]. Among them, the *OsDXS2* belongs to type of II-DXS enzymes and plays a vital role as a speed limited enzyme providing IPP/DMAPPs to the accumulation of carotenoid in rice seeds [28]. In this study, different from previously reported genes, *TSCD5/OsDXS1* belonged to I-DXS type enzymes, and its mutation caused early albino leaves and abnormal chloroplast at high

temperature in rice. In addition, the transcript levels of genes related to chlorophyll biosynthesis, photosynthesis and chloroplast development were significantly down-regulated. In addition, similar to the *hsa1* and *tscd11* mutants, compared with WYJ7, the transcript level of *TSCD5* in *tscd5* was significantly lower at 25 °C and 35 °C, and even lower in *tscd5* plants cultured at 35 °C. The results indicated that the expression level of *TSCD5* in *tscd5* mutant was inhibited, at same time high temperature enhanced the inhibition. In addition, the lethal albinism of *tscd5* mutant may be caused by the inhibition of chloroplast development under high temperature (Figure 3D,H). Similar to *TCM5* and *HSA1*, TSCD5 protein is also localized in chloroplasts, confirming that *TSCD5* gene is necessary for chloroplast development under high temperature.

Many biotic and abiotic stresses, such as heat, cold, drought, salt, and wounding, caused chloroplast damage in plants [57–62]. TEM observation showed that under high temperature, chloroplast development of *tscd5* were hampered (Figure 3D,H). At 25 °C, pigments contents and transcript levels of genes associated with Chl biosynthesis, photosynthesis and chloroplast development had no significant difference between WYJ7 and *tscd5.* In contrast, pigments contents and transcription level of these genes are seriously affected in *tscd5* at 35 °C. Interestingly, the chloroplasts of *tscd5*, appearing different from those of the WT when grown at 35 °C, were almost similar to those of WYJ at 25 °C. The findings indicated that *tscd5* was more sensitive to high temperature than WYJ7. The *tscd5* mutant died, maybe due to the incomplete development of chloroplast, and even the loss of normal function of chloroplast under high temperature.

As the chloroplast induced damage, plants tended to grow in weakness. These weak plants are often sensitive to ROS, especially in stress situations [9,63]. Common stress situations include high light or high temperature. ROS accumulation led to DNA damage, lipid peroxidation, oxidative damage to thylakoid membranes and other cellular components [64–66]. In this study, at 25 °C, there is no significant difference in the content of $H_2O_2$ and MDA and activities of CAT, SOD and POD activity between wild types. However, $H_2O_2$ and MDA contents were increased in the *tscd5* mutant compared to theWYJ7 at 35 °C, which is corresponded to the results of NBT and DAB staining. Under oxidative stress situation, plants can synthesize anti-oxidative enzymes, such as SOD and CAT, to eliminate ROS [67,68]. SOD catalyzes the dismutation of $O^{2-}$ to produce $H_2O_2$, and CAT is the major $H_2O_2$-scavenging enzyme. Reasonably, the SOD activity was higher and CAT activity was lower in *tscd5* than WYJ7 at 35 °C (Figure 4). Treatment with exogenous antioxidant AsA eased the local albino phenotype of *tscd5* seedlings grown at 35 °C (Supplementary Figure S4), demonstrating that the excess ROS accumulated at 35 °C impaired the oxidative balance in the *tscd5* mutant. The expression levels of most genes associated with antioxidant, such as *AOX1a*, *AOX1b*, *APX1* and *CATB* were significantly induced in the *tscd5* mutant (Supplementary Figure S2C,D). Although RT-qPCR results showed that their expression levels of most ROS-scavenging genes were significantly increased in the *tscd5* mutant at 35 °C, it may not be adequate to reduce ROS to the level of WYJ7 plants.

## 5. Conclusions

In summary, it can be concluded that the *TSCD5* gene, encoding a 1-deoxy-D-xylulose-5-phosphate synthase 1 in rice was essential for early chloroplast development under high temperature and its mutation would lead to an aberrant chloroplast and abnormal transcript levels of genes associated with chlorophyll biosynthesis, photosynthesis and chloroplast development under heat stress. These results provide a basis for further study of the molecular mechanism for chloroplast development under high temperature.

**Supplementary Materials:** The following supporting information can be downloaded at: https://www.mdpi.com/article/10.3390/agriculture13030563/s1, Supplementary Figure S1 Transmission electron microscopy observation of chloroplasts in WYJ7 and the *tscd5* leaves. (A,B) Chloroplasts structure from WT and *tscd5* at tillering stage under natural high-temperature field conditions (Scale bar = 1 m). (C,D) are enlargements of A-B, respectively (Scale bar = 0.5 μm). CP, chloroplast;

G, granum; OG, osmiophilic granule. Supplementary Figure S2 Alteration in expression level of senescence and ROS related genes in *tscd5*. (A,B) Expression of genes associated with senescence (A) and ROS (B) in WYJ7 and *tscd5* mutant at 25 °C. (C,D) Expression of genes associated with senescence (C) ROS (D) in WYJ7 and *tscd5* mutant at 35 °C. *Histone* was used as a control for qRT-PCR. The expression level of each tested genes in WYJ7 was set to 1.0. Mean ± SD, n = 3, ** extremely significance at $p < 0.01$ (Student's *t*-test). Supplementary Figure S3 Sequence alignment and phylogenetic analysis of TSCD5. (A) Alignment of TSCD5 protein homolgs from 9 plant species. Blue or pink shades indicate fully or partially conserved amino acid. (B) Phylogenetic tree of TSCD5. Protein sequences include *Oryza sativa Japonica* Group (OsTSCD5, XP_015640505.1), *Panicum miliaceum* (PmTSCD5, RLN01033.1), *Zea mays* (ZmTSCD5, ACG27905.1), *Brachypodium distachyon* (BdTSCD5, XP_003568467.1), *Setaria italica* (SiTSCD5, XP_004962111.1), *Triticum turgidum subsp. durum* (TtTSCD5, VAH07338.1), *Poa pratensis* (PpTSCD5, AXS78096.1), *Aegilops tauschii subsp. tauschii* (AtTSCD5, XP_020161542.1), *Sorghum bicolor* (SbTSCD5, XP_002441088.1). Supplementary Figure S4 Effect of exogenous ascorbic acid (AsA) on the *tscd5* mutant. (A, B) The phenotype changes in new leaves of *tscd5* mutant seedlings that were untreated or pretreated with an exogenous application of 1mM AsA when they were transferred from the 25 °C to the 35 °C condition at the 2-leaf stage. Scale bar = 2 cm. (A) CK, control check; (B) 1 mM AsA. Supplementary Table S1 Comparison of agronomic traits between wild type WYJ7 and mutant *tscd5*. Mean ± SD, n = 9, * significance at $p < 0.05$, ** extremely significance at $p < 0.01$ (Student's *t*-test). Supplementary Table S2 Genetic analysis of the *tscd5* mutant in F$_2$ population. Supplementary Table S3 Primers for fine mapping in this study. Supplementary Table S4 Primers for vector construction in this study. Supplementary Table S5 Primers for qRT-PCR in this study.

**Author Contributions:** S.Y. and G.F. performed the experiments and wrote the manuscript; Z.G. supervised this research; S.Y., G.F., B.R., A.Z., Y.Z., G.Y., W.S. and H.G. performed the experiments and analyzed the data. G.F., B.R., J.W. and Z.G. revised the manuscript. All authors read and approved the final manuscript.

**Funding:** This work was supported by the National Natural Science Foundation of China (Grant No. 31901481), the Central Public-interest Scientific Institution Basal Research Fund (NO.CPSIBRF-CNRRI-202111) and the China Agriculture Research System (No.CARS-09).

**Informed Consent Statement:** All co-authors involved in the paper consent to publish this article in Agriculture.

**Data Availability Statement:** All relevant data are provided as figures or table within the paper.

**Conflicts of Interest:** The authors declare no conflict of interest.

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
