# Peer review of "Identification of Thermo-Sensitive Chloroplast Development Gene TSCD5 Required for Rice Chloroplast Development under High Temperature"

_agriculture, doi:10.3390/agriculture13030563_

Round 1

Reviewer 1 Report

The work on ‘Identification of thermo-sensitive chloroplast development 2 gene TSCD5 required for rice chloroplast development under high temperature’ is a very good study. Before acceptance of the manuscript; the following points may be confirmed:

1.       Is the light intensity (200 μmol m-2 s -1) under green house for the study is adequate? Please provide the reference for the use and incorporate in the manuscript.

2.       Is the linked to any of the tms genes in rice? If not why to be used for markers in hybrid breeding[5] (Line 40 under introduction)?

3.       Few language mistakes exist in the manuscript. Please improve the writing of the manuscript for language.

Author Response

Response to Reviewer 1 Comments

The work on ‘Identification of thermo-sensitive chloroplast development 2 gene TSCD5 required for rice chloroplast development under high temperature’ is a very good study. Before acceptance of the manuscript; the following points may be confirmed:

1.Is the light intensity (200 μmol m-2 s -1) under green house for the study is adequate? Please provide the reference for the use and incorporate in the manuscript.

R1: Thank you for suggestion. Light intensity under greenhouse (200 μmol m-2 s -1) is sufficient, such as for tscd5 and ls1 mutants, and we have provided corresponding references ([9, 18])

2.Is the linked to any of the tms genes in rice? If not why to be used for markers in hybrid breeding [5] (Line 40 under introduction)?

R2: Good suggestion. TSCD5 controls quality traits, so it cannot be used for markers in hybrid breeding. We have deleted the sentence in Introduction.

3.Few language mistakes exist in the manuscript. Please improve the writing of the manuscript for language.

R3: Thanks. We have further checked and revised the language of the manuscript.

Reviewer 2 Report

Yang et al conducted a study to underline the role of TSCD5 gene in chloroplast development in rice plants. The manuscript provides most of the required information and can be considered for acceptance, however, there are the following crucial points that can be considered by authors to increase the value of the manuscript and maybe readability.

- English language needs to be corrected. The manuscript can be read and corrected by a native English speaker. For example, Line 27-29. Line 33-‘are’ is missing. Line 68. Line 74-‘c’ of chloroplasts should be small. Line 101. Line 104-sentence should not start with ‘AND’. Line 208 etc.

- In the method section, please add a separate section for statistical analysis and add the statistical method  followed for all the measurements. Also, name the program used for the analysis.

-Discussion must be elaborated to discuss the results of each measurement. For example, MDA results are not discussed. There are so many studies discussing the association between photosynthetic measurement and antioxidant enzymes. Include discussion from the following studies and cite them (10.3389/fpls.2021.736614; 10.1007/s13205-020-02479-9)

- Please mention ‘rice’ in every figure caption. For example, Figure 2, Figure 4.

I believe that the manuscript can be accepted once the authors address the mentioned points and enrich the manuscript with crucial information. 

Author Response

Response to Reviewer 2 Comments

Comments and Suggestions for Authors

Yang et al conducted a study to underline the role of TSCD5 gene in chloroplast development in rice plants. The manuscript provides most of the required information and can be considered for acceptance, however, there are the following crucial points that can be considered by authors to increase the value of the manuscript and maybe readability.

- English language needs to be corrected. The manuscript can be read and corrected by a native English speaker. For example, Line 27-29. Line 33-‘are’ is missing. Line 68. Line 74-‘c’ of chloroplasts should be small. Line 101. Line 104-sentence should not start with ‘AND’. Line 208 etc.

R1: Thank you for suggestion. The manuscript has been corrected by a native English speaker.

- In the method section, please add a separate section for statistical analysis and add the statistical method followed for all the measurements. Also, name the program used for the analysis.

R2: In the method section, we have added a separate section for statistical analysis and add the statistical method followed for all the measurements in 2.14.

-Discussion must be elaborated to discuss the results of each measurement. For example, MDA results are not discussed. There are so many studies discussing the association between photosynthetic measurement and antioxidant enzymes. Include discussion from the following studies and cite them (10.3389/fpls.2021.736614; 10.1007/s13205-020-02479-9)

R3: Good suggestion. At 25℃, there is no difference in the content of H2O2 and MDA and activities of CAT, SOD and POD activity between wild types. However, H2O2 and MDA contents were increased in the tscd5 mutant compared to theWYJ7 at 35℃, which is corresponded to the results of NBT and DAB staining. And we have added discussion on MDA results in Discussion and referred to some studies in red. 

- Please mention ‘rice’ in every figure caption. For example, Figure 2, Figure 4.

R4: We have added ‘rice’ in every figure caption and marked it in red.

I believe that the manuscript can be accepted once the authors address the mentioned points and enrich the manuscript with crucial information. 

Reviewer 3 Report

In this paper the authors identify a rice mutant (tscd5) that shows albinism when grown at high temperature. They show that tscd5 mutants are impaired in chloroplast development at high temperature, shows defective isoprenoid accumulation, have increased ROS accumulation, and higher amounts of cell death. Using RT-qPCR, the authors identify gene expression signatures of several of these phenotypes. A particular role for excess ROS accumulation in the phenotype is nicely demonstrated by phenotypic rescue at the non-permissive temperature by application of ascorbic acid. The authors go on to map and transgenically complement the casual EMS-induced variant in an F2 population and find that a point mutation in a 1-deoxy-D-xylulose-5-phosphate synthase encoding gene is responsible. The encoded protein is chloroplast localized and broadly expressed throughout the plant and downregulated under high temperature. 

Overall the presentation of the results is quite compelling but the manuscript would benefit if the authors addressed the following:

The discussion would benefit if the authors better addressed the connection between ROS and the biochemical outputs of the pathway that TSCD5 acts in. It seems to me that the cell death in tscd5 mutants happens because of a lack of ROS-scavenging carotenoids and other isoprenoid compounds but this is never directly addressed? 

Furthermore, a more in-depth discussion of how the phenotypes presented in this study compare to what's known about TSCD5 mutants in other species would be useful. For example, the authors state that cla1/chs5 mutants show a chlorotic phenotype at low temperature, which is the opposite of what's shown here. This could also lead to a discussion regarding whether other stresses aside from heat might illicit phenotypes in tscd5 mutants. 

Is the EMS variant expected to be a complete loss-of-function allele? Is it in a highly conserved region or domain known to be important for enzymatic function?

Finally, one issue with the data presentation is mislabelling of figure panels in the figure legends. For example, the legend for Fig. 1C,D,E and 1F,G do not match the panels. Same issue with Fig. 1H-L, Fig. 2I-L. It is also unclear which panels correspond to mutant and WT in Figure 3I-P. 

Minor comments:

The chloroplasts in tscd5 mutants at 25C appear to have lighter material or spaces between the grana lamella stacks (Fig. 3B,F). Is this a consistently seen phenotype? 

It would be helpful if the authors included bright field images in Fig. 3I-P as it is difficult to see what the tissue context being shown is.

Citations (even if just review or single RNA-Seq study) are needed for marker genes used in the gene expression analyses. 

Author Response

Response to Reviewer 3 Comments

Comments and Suggestions for Authors

In this paper the authors identify a rice mutant (tscd5) that shows albinism when grown at high temperature. They show that tscd5 mutants are impaired in chloroplast development at high temperature, shows defective isoprenoid accumulation, have increased ROS accumulation, and higher amounts of cell death. Using RT-qPCR, the authors identify gene expression signatures of several of these phenotypes. A particular role for excess ROS accumulation in the phenotype is nicely demonstrated by phenotypic rescue at the non-permissive temperature by application of ascorbic acid. The authors go on to map and transgenically complement the casual EMS-induced variant in an F2 population and find that a point mutation in a 1-deoxy-D-xylulose-5-phosphate synthase encoding gene is responsible. The encoded protein is chloroplast localized and broadly expressed throughout the plant and downregulated under high temperature. 

Overall the presentation of the results is quite compelling but the manuscript would benefit if the authors addressed the following:

The discussion would benefit if the authors better addressed the connection between ROS and the biochemical outputs of the pathway that TSCD5 acts in. It seems to me that the cell death in tscd5 mutants happens because of a lack of ROS-scavenging carotenoids and other isoprenoid compounds but this is never directly addressed? 

R1: Thank you for suggestion. Actually, the tscd5 mutant died may due to the incomplete development of chloroplast, and even the loss of normal function of chloroplast under high temperature. And increased accumulation of reactive oxygen species (ROS) usually occurs in weak plants under high temperature. At 25℃, there is no significant difference in the content of H2O2 and MDA and activities of CAT, SOD and POD activity between wild types. However, H2O2 and MDA contents were increased in the tscd5 mutant compared to theWYJ7 at 35℃, which is corresponded to the results of NBT and DAB staining. Besides, we have added these in discussion in red.

Furthermore, a more in-depth discussion of how the phenotypes presented in this study compare to what's known about TSCD5 mutants in other species would be useful. For example, the authors state that cla1/chs5 mutants show a chlorotic phenotype at low temperature, which is the opposite of what's shown here. This could also lead to a discussion regarding whether other stresses aside from heat might illicit phenotypes in tscd5 mutants. 

R2: Good suggestion. Both high temperature and low temperature are adverse conditions, and the cla1/chs5 mutants also have damaged chloroplasts and chlorosis phenotype under low temperature, which is similar to tscd5 mutants’ phenotype. In addition, evolutionary tree research shows that the evolutionary relationship between Arabidopsis and rice is far away, which is not shown in Supplementary Figure 3. Therefore, it is normal to have different functions, that is, different phenotypes at different temperatures. And there may be other conditions that can cause chloroplast damage, and similar phenotype may also occur.

Is the EMS variant expected to be a complete loss-of-function allele? Is it in a highly conserved region or domain known to be important for enzymatic function?

R3: At normal temperature, chloroplast can maintain almost normal function despite gene mutation. Therefore, at 25℃, the phenotype of the mutant was similar to that of the wild type. However, under high temperature, the chloroplast function was almost completely lost, resulting in the death of the mutant. Protein sequences alignment showed that TSCD5 were highly conserved in many plant species (Supplementary Figure S3A). Comparison of three-dimensional modes of TSCD5 from WYJ7 and tscd5 suggested that although little difference existed in TSCD5 structure, the number of hydrogen bond between peptides was increased in TSCD5 of tscd5.

Finally, one issue with the data presentation is mislabelling of figure panels in the figure legends. For example, the legend for Fig. 1C,D,E and 1F,G do not match the panels. Same issue with Fig. 1H-L, Fig. 2I-L. It is also unclear which panels correspond to mutant and WT in Figure 3I-P. 

R4: Thanks. We have modified the label of the figure panels in the figure legends.

Minor comments:

The chloroplasts in tscd5 mutants at 25C appear to have lighter material or spaces between the grana lamella stacks (Fig. 3B,F). Is this a consistently seen phenotype? 

R5: At normal temperature, chloroplast can maintain almost normal function despite gene mutation. Therefore, at 25℃, the phenotype of the mutant was similar to that of the wild type.

It would be helpful if the authors included bright field images in Fig. 3I-P as it is difficult to see what the tissue context being shown is.

R6: The leaves at the trefoil stage are tender, and the tissue that can be observed is small, so the background can only reach this brightness. The difference between the wild type and the mutant can be seen from this picture.

Citations (even if just review or single RNA-Seq study) are needed for marker genes used in the gene expression analyses. 

R7: We have added the Histone gene as the internal reference (marker gene) in 2.13 in red and cited a reference[9].

Round 2

Reviewer 2 Report

The authors have implemented the suggestions.